# SCHEME: SCALABLE CHANNEL MIXER FOR VISION TRANSFORMERS

## ABSTRACT

Vision Transformers have achieved impressive performance in many vision tasks. While the token mixer or attention block has been studied in great detail, much less research has been devoted to the channel mixer or feature mixing block (FFN or MLP), which accounts for a significant portion of the model parameters and computation. In this work, we show that the dense MLP connections can be replaced with a block diagonal MLP structure that supports larger expansion ratios by splitting MLP features into groups. To improve the feature clusters formed by this structure we propose the use of a lightweight, parameter-free, channel covariance attention (CCA) mechanism as a parallel branch during training. This enables gradual feature mixing across channel groups during training whose contribution decays to zero as the training progresses to convergence. As a result, the CCA block can be discarded during inference, enabling enhanced performance at no additional computational cost. The resulting *Scalable CHannEl MixEr* (SCHEME) can be plugged into any ViT architecture to obtain a gamut of models with different trade-offs between complexity and performance by controlling the block diagonal MLP structure. This is shown by the introduction of a new family of SCHEMEformer models. Experiments on image classification, object detection, and semantic segmentation, with different ViT backbones, consistently demonstrate substantial accuracy gains over existing designs, especially for lower complexity regimes. The SCHEMEformer family is shown to establish new Pareto frontiers for accuracy vs FLOPS, accuracy vs model size, and accuracy vs throughput, especially for fast transformers of small size.

## 1 INTRODUCTION

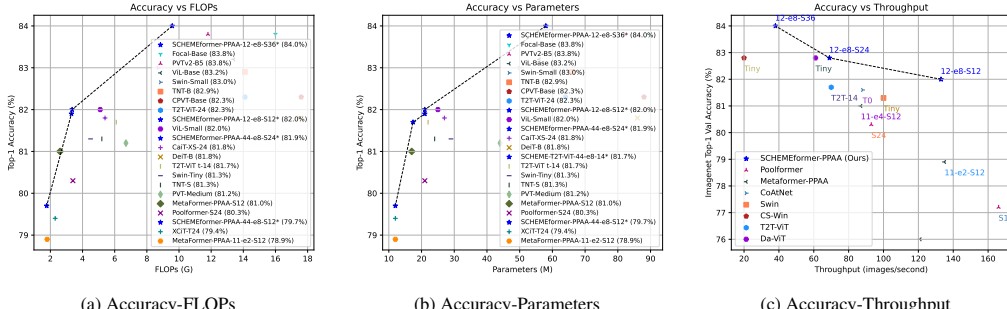

(a) Accuracy-FLOPs      (b) Accuracy-Parameters      (c) Accuracy-Throughput

Figure 1: Comparison of the proposed SCHEMEformer family, derived from the Metaformer-PPAA-S12 model (52) with higher expansion ratios in the MLP blocks, and many SOTA transformers from the literature. The SCHEMEFormer family establishes a new Pareto frontier (optimal trade-off) for a) accuracy vs. FLOPs, b) accuracy vs model size, and c) accuracy vs, throughput. SCHEMEformer models are particularly effective for the design of fast transformers (throughput between 75 and 150 images/s) with small model size.See 5 for zoomed version.

Vision Transformers (ViTs) (11; 26; 42; 25) are now ubiquitous in computer vision. They decompose an image into a set of patches which are fed as tokens to a transformer model (41) of two

main components: a *spatial attention* module, which reweighs each token according to its similarity to the other tokens extracted from the image, enabling information fusion across large spatial distances, and a *channel mixer* that combines the feature channels extracted from all patches using a multi-layer perceptron (MLP or FFN). A bottleneck of this model is the quadratic complexity of the attention mechanism on the number of patches. Numerous ViT variants have been proposed to address the problem, using improved attention mechanisms or hybrid architectures that replace attention or combine it with convolutions. Much less research has been devoted to the channel mixer. Most models simply adopt the two-layer MLP block of (41), where channels are first expanded by a specified *expansion ratio* and then compressed to the original dimension. This is somewhat surprising since the mixer is critical for good transformer performance. For example, it is known that 1) pure attention, without MLPs or residual connections, collapses doubly exponentially to a rank one matrix (10), 2) training fails to converge without residual connections or MLP (52), and 3) replacing MLPs with more attention blocks (both spatial and channel attention) of equivalent computational complexity lowers the transformer accuracy (8). All these observations indicate that the channel mixer is an indispensable ViT component.

In this work, quantify how much the channel mixer module contributes to ViT performance and investigate how to improve the trade-off between complexity and accuracy of the ViT model. We show that enhanced design of the channel mixer can lead to significant improvements in transformer performance by introducing a novel *Scalable CHannEl MixEr* (SCHEME) that enables the design of models with larger expansion ratios. SCHEME is a generic channel mixer that can be plugged into existing ViT variants to obtain effective scaled-down or scaled-up model versions. We replace the channel mixer of a state of the art architecture (SOTA) for low-complexity transformers, the MetaFormer-PPAA-S12 (52), with SCHEME to obtain a new family of **SCHEMEformer** models with improved accuracy/complexity trade-off. This is illustrated in Figure 1, where the SCHEMEformer family is demonstrated to establish new Pareto frontiers for accuracy vs FLOPS, accuracy vs model size, and accuracy vs throughput, showing that SCHEME allows fine control over all these variables, while guaranteeing SOTA performance. These properties are shown to hold for image classification, object detection and semantic segmentation tasks, as well as for different architectures, such as T2T-ViT (54), CoAtNet (5), Swin Transformer (25), CSWin (9) and DaViT (8).

To develop SCHEME, we start by studying the impact of the mixer channel expansion. Transformer performance is shown to increase with expansion ration until it saturates for an expansion ratio beyond 8. However, because mixer (MLP) complexity also increases with the dimension of the intermediate representation, naive channel scaling with expansion ratios larger than 4 causes an explosion of parameters and computation, leading to models of large complexity and prone to overfit. To achieve a better trade-off between dimensionality and computation, we leverage recent findings about the increased hardware efficiency of mixers with block diagonal structure (1; 6). These group the input and output feature vectors of a layer into disjoint subsets and perform matrix multiplications only within each group, as illustrated in Figure 2. We denote the resulting MLP block as *Block Diagonal MLP* (BD-MLP). Despite the lack of feature mixing across groups, we find that a transformer model equipped with the BD-MLP and a larger expansion ratio (to match both the parameter count and computation) achieves comparable or slightly higher accuracy than a baseline with dense MLP. This suggests that the lower accuracy of the block diagonal operations is offset by the gains of larger expansion ratios. Further analyzing the features learned by the different groups of the BD-MLP, we observe that they form feature clusters of similar ability to discriminate the target classes. To learn better feature clusters, we seek a mechanism capable of restoring inter-group feature communication during training without increasing parameters. For this, we propose a channel attention branch that reweighs the input features of the BD-MLP according to their covariance matrix, as illustrated in Figure 2. This attention mechanism is denoted as the *channel covariance attention* (CCA) block. The re-weighted features are then fused with the BD-MLP output by means of a weighted residual addition with learned weights $(\alpha, 1 - \alpha)$.

As shown in Figure 2, SCHEME combines the sparse block diagonal structure of the BD-MLP, and the parameter-free CCA attention module, to implement a channel mixer extremely efficient in terms of parameters. Ablations of the evolution of the fusion weight $1 - \alpha$ learned in the CCA branch (see Fig. 2) over training show that it gradually decays to zero during training. This happens consistently across all layers of the model and across model architectures. Hence, while the CCA is important for the formation of good feature clusters during training, it can be removed at inference without any loss, as illustrated in Table 3 and Figure 2. As a result, the model accuracy improves over simply

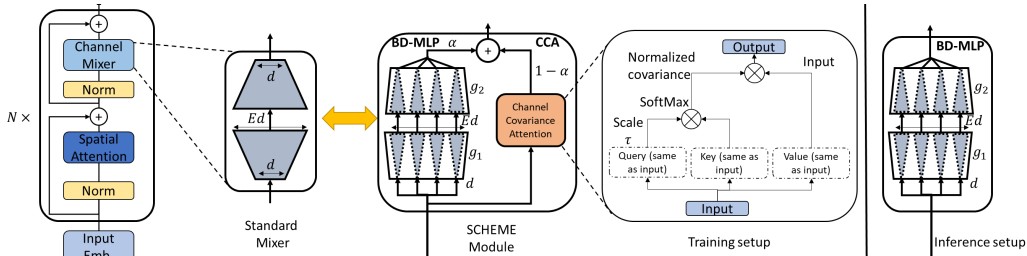

Figure 2: Proposed SCHEME channel mixer. The channel mixer of the standard transformer consists of two MLP layers, performing dimensionality expansion and reduction by a factor of $E$. SCHEME uses a combination of a block diagonal MLP (BD-MLP), which reduces the complexity of the MLP layers by using block diagonal weights, and a channel covariance attention (CCA) mechanism that enables communication across feature groups through feature-based attention. This, however, is only needed for training. The weights $1 - \alpha$ decay to zero upon training convergence and CCA can be discarded during inference, as shown on the right. Experiments show that CCA helps learn better feature clusters, but is not needed once these are formed.

using the BD-MLP mixer, but inference complexity does not. This leads to an extremely efficient inference setup, both in terms of parameters and FLOPs. Overall, the paper makes the following contributions,

- a study of the channel mixer of ViT MLPs, showing that dense feature mixing can be replaced by sparse feature mixing of higher internal feature dimensionality for improved accuracy, without increased complexity.

- the SCHEME module, which combines 1) a BD-MLP to enable internal feature representations of larger dimensionality than previous MLP blocks, and 2) CCA to enable the learning of these representations without cost at inference.

- various models that combine SCHEME with previous transformer architectures to achieve SOTA trade-offs between accuracy and model size, FLOPS, or throughput, such as the SCHEMEformer of Figure 1. This is shown particularly effective for the design of fast transformers with small model size, of interest for edge devices, robotics, and low-power applications.

- Experiments on image classification, object detection, and semantic segmentation, showing consistent gains in accuracy for fixed computation and size.

## 2  RELATED WORK

**Vision Transformers:** Vision transformers advanced the SOTA in several vision tasks since (26; 11) successfully applied the transformer-based self-attention NLP model of (41) to image generation and classification tasks. These models rely on a spatial attention mechanism, based on the matrix of dot-products between features extracted from image patches. This has quadratic complexity in the number of patches and is quite intensive. Most follow up work (15; 33; 53; 38; 34; 46; 35; 54; 55; 40) improved the spatial attention mechanism of ViT. DeiT (35) and subsequent works (36; 37) introduced a distillation token to distill information, typically from a CNN teacher, without large amounts of data or compute. PvT (42) proposed a progressive shrinking pyramid architecture with spatial-reduction attention that scales ViTs for dense prediction tasks beyond image classification. Swin transformers (25) introduced a hierarchical shifted window attention mechanism, which reduces the complexity to linear with respect to the number of windows. HaloNet (40) proposed two extensions for local ViTs (27) with blocked local attention and relaxed translational equivariance for scaling ViTs. A more extensive review of ViTs is given in (24). Recently, several works have shown that spatial attention is not the critical ViT feature. Some works improved performance by relying on hybrid architectures, which augment or replace ViT layers with convolutions (45; 39). Efficient transformer designs, for edge devices, frequently sacrifice spatial attention to achieve better trade-offs between FLOPs and accuracy (2; 21). Other works have questioned the need for spatial attention altogether. Metaformer (52) argued that the fundamental trait of ViT is the mixing of information across patches, showing that competitive results can be obtained by simply replacing attention with pooling or identity operations. Similarly, (23) showed that spatial attention is not critical for vision transformers by proposing a spatial-gating MLP of comparable performance to ViT (11). DaViT (8) showed that spatial attention is helpful by reusing its design for channel attention, building a cascade

| Model | Ratio (E) | FLOPs (G) | Top-1 Acc (%) |
|---|---|---|---|
| Metaformer* | 8 | 4.14 | 81.6 (+0.6) |
| Metaformer* | 6 | 3.35 | 81.8 (+0.8) |
| Metaformer (52) | 4 | 2.55 | 81.0 (+0.0) |
| Metaformer* | 2 | 1.77 | 78.9 (-2.1) |
| Metaformer* | 1 | 1.33 | 76.0 (-5.0) |
| Metaformer-S18* | 1 | 2.51 | 78.3 (-2.7) |

Table 1: MetaFormer-S12 (52) ImageNet-1K validation accuracy vs MLP expansion ratio ($E$). *: results by author code.

| | SCHEME | |
|---|---|---|
| | 44-e8 | 12-e8 |
| S12 P (M) | 12 | 21 |
| S12 F (G) | 1.8 | 3.3 |
| S24 P (M) | 21 | 40 |
| S24 F (G) | 3.3 | 6.5 |
| S36 P (M) | 55 | 59 |
| S36 F (G) | 8.0 | 9.6 |

Table 2: SCHEME parameters (P) and FLOPs (F). 44-e8 (12-e8) downscales (upscale) the MetaFormer model.

| SCHEME | CCA Used | Params (M) | FLOPs (G) | Top-1 Acc (%) |
|---|---|---|---|---|
| 44-e8-S12 | ✓ | 11.83 | 2.16 | 79.74 |
| 44-e8-S12 | | 11.83 | 1.77 | 79.72 |
| 12-e8-S24 | ✓ | 40.0 | 7.3 | 82.80 |
| 12-e8-S24 | | 40.0 | 6.5 | 82.76 |
| CoatNet-44-e8 | ✓ | 17.80 | 3.83 | 80.70 |
| CoatNet-44-e8 | | 17.80 | 3.42 | 80.68 |
| Swin-12-e8-T | ✓ | 36.93 | 7.00 | 81.69 |
| Swin-12-e8-T | | 36.93 | 5.89 | 81.69 |

Table 3: Impact of removing CCA branch during inference.

of alternating spatial and channel attention blocks. While this improves performance, it increases the complexity of the transformer block, resulting in a model with many parameters and potentially redundant channel mixer operations. Despite all this work on ViT architectures, little emphasis has been devoted to the channel mixer module (MLP) that follows attention. This is surprising because the mixer dominates both the parameter count and complexity (FLOPs) of the standard transformer block. (22; 53) modify the MLP block to mimic the inverted residual block of the MobileNetV2 (29), by adding a depthwise convolution. This improved performance but increases parameter and computation costs. Switch Transformer (13) replaces the FFN with a sparse mixture of experts (32) that dynamically routes the input tokens. This design allows scaling models to large sizes using higher number of experts but is not effective for ViTs. XCiT (12) employs a cross-covariance attention (XCA) operator, which can be seen as a "transposed" version of self-attention that operates across feature channels. The architecture of XCiT is composed of three primary components: the core XCA operation, a local patch interaction (LPI) module, and a feedforward network (FFN). The XCA mechanism computes the covariance operation across different head groups, akin to multi-head attention. In this work, we extend this concept by proposing a Cross-Covariance Channel Attention (CCA) operation to facilitate feature mixing across different channel groups. Unlike XCiT, which uses "heads" for interaction and projection matrices for queries, keys and values, CCA leverages the full feature set to compute covariance without any projections, ensuring a more comprehensive representation of inter-feature interactions. In this work, we propose an efficient and generic channel mixer module (BD-MLP and CCA) that improves both the parameter and computational efficiency of the transformer and allows for flexible scaling of ViTs.

## 3 THE SCHEME MODULE

Figure 2 depicts the proposed SCHEME module for feature mixing in ViTs. As shown on the left, the standard channel mixer consists of two MLP layers, which expand the dimensionality of the input features and then reduce it to the original size. Let $\mathbf{x} \in \mathbb{R}^{d \times N}$ be the matrix containing the $N$ $d$-dimensional input feature vectors extracted from $N$ image patches. The mixer computes an intermediate representation $\mathbf{z} \in \mathbb{R}^{Ed \times N}$ and an output representation $\mathbf{y} \in \mathbb{R}^{d \times N}$ according to

$$\mathbf{z} = \sigma(\mathbf{W}_1 \mathbf{x} + \mathbf{b}_1 \mathbf{1}_N^T) \tag{1}$$

$$\mathbf{y} = \mathbf{W}_2 \mathbf{z} + \mathbf{b}_2 \mathbf{1}_N^T \tag{2}$$

where $\mathbf{W}_1 \in \mathbb{R}^{Ed \times d}, \mathbf{W}_2 \in \mathbb{R}^{d \times Ed}, \mathbf{b}_1 \in \mathbb{R}^{Ed}, \mathbf{b}_2 \in \mathbb{R}^d$, $\mathbf{1}_N$ is the $N$-dimensional vector containing ones as all entries, $\sigma(.)$ is the activation function (typically GELU), and $E$ is an expansion factor, typically 4.

Table 1 details the impact of the mixer on overall transformer performance by evaluating the role of the expansion factor $E$ on the performance of Metaformer-PPAA-S12 (52) (Pooling, Pooling, Attention, Attention) architecture on ImageNet-1K. Classification accuracy increases from 76.0 to 81.8%, as $E$ ranges from 1 to 6, decreasing for $E = 8$, which suggests overfitting. The table also shows that these gains are not trivial. The S18 model, which has no expansion ($E = 1$) but more transformer layers and complexity comparable to that of the S12 model with $E = 4$, has an accuracy 2.7 points lower than the latter. In summary, for a given computation budget, it is beneficial to trade off transformer depth for dimensionality expansion in the channel mixer. This shows that this expansion is a critical component of the transformer architecture. On the other hand, naively scaling $E$ beyond 6 severely increases parameters and computation, leading to models that over-fit and are impractical for many applications.

## 3.1 SCALABLE CHANNEL MIXER (SCHEME)

**Block Diagonal MLP (BD-MLP):** Block diagonal matrices have been previously used to efficiently approximate dense matrices (1; 6). In CNNs, group channel operations are frequently used to design lightweight mobile models with improved accuracy-computation trade-off (19; 3; 29). This consists of splitting the feature vectors of (1)-(2) into disjoint groups, e.g. $\mathbf{x}$ into a set of $g_1$ disjoint features $\{\mathbf{x}_k\}_{k=1}^{g_1}$ where $\mathbf{x}_k \in \mathbb{R}^{d/g_1 \times N}$, and $\mathbf{y}$ into a set $\{\mathbf{y}_k\}_{k=1}^{g_2}$ where $\mathbf{y}_k \in \mathbb{R}^{Ed/g_2 \times N}$. As illustrated in Figure 2, the MLPs of (1)-(2) are then implemented independently for each group, according to

$$\mathbf{z}_k = \sigma(\mathbf{W}_{1,k}\mathbf{x}_k + \mathbf{b}_{1,k}\mathbf{1}_N^T) \tag{3}$$

$$\mathbf{y}_k = \mathbf{W}_{2,k}\mathbf{z}_k + \mathbf{b}_{2,k}\mathbf{1}_N^T \tag{4}$$

where $\mathbf{W}_{1,k} \in \mathbb{R}^{Ed/g_1 \times N/g_1}, \mathbf{W}_{2,k} \in \mathbb{R}^{d/g_2 \times Ed/g_2}, \mathbf{b}_1 \in \mathbb{R}^{Ed/g_1}, \mathbf{b}_2 \in \mathbb{R}^{d/g_2}$ and $\mathbf{z}$ is decomposed into a set $\{\mathbf{z}_k\}_{k=1}^{G}$ where $\mathbf{z}_k \in \mathbb{R}^{Ed/G \times N}$, with $G = g_1$ in (3) and $G = g_2$ in (4). Since the complexity of (3) is $g_1^2$ times smaller than that of (1) and there are $g_1$ groups, the complexity of the first MLP is $1/g_1$ times that of standard MLP. Similarly, the complexity of the second MLP is $1/g_2$ times that of the standard MLP. Hence, a transformer equipped with the BD-MLP and expansion factor $\frac{2g_1g_2}{g_1+g_2}E$ has identical complexity to a standard transformer of factor $E$. For example, when $g_1 = g_2 = g$ this allows growing the expansion factor by a factor of $g$ without computational increase.

**Channel Covariance Attention (CCA):** While the introduction of groups enables accuracy gains due to the increased expansion factor by $\frac{2g_1g_2}{g_1+g_2}$, it results in sub-optimal features. This is because the features in the different groups of (3)-(4) are processed *independently*, i.e. there is no inter-group feature fusion. This reduces the efficiency of the BD-MLP. To enable feature mixing between *all* feature channels and thus induce the formation of better feature clusters, we introduce a covariance attention mechanism in a parallel branch, as illustrated in Figure 2. The input features are first transposed to obtain the $d \times d$ covariance matrix[1] $\mathbf{x}\mathbf{x}^T$. This is then used to re-weigh the input features by their covariance with other feature channels, using

$$CCA(\mathbf{x}) = \text{softmax}\left(\frac{\mathbf{x}\mathbf{x}^T}{\tau}\right)\mathbf{x} \tag{5}$$

where the softmax operation is applied across the matrix rows and $\tau$ is a smoothing factor. The output of the channel mixer block is the weighted summation of the BD-MLP and CCA branches

$$\mathbf{y}_{out} = \alpha\mathbf{y} + (1 - \alpha)CCA(\mathbf{x}), \tag{6}$$

where $\alpha$ is a mixing weight learned across all samples. Various other design choices are discussed in Section 4.2.

**CCA as a Regularizer:** The introduction of a parameter free attention branch and a learnable weight $\alpha$ allows the model to form better feature clusters during training and gradually decay the contribution from CCA branch once the feature clusters are formed. This can be seen in Figure 3, which plots the value of the learned mixing weight $1 - \alpha$ as a function of training epochs on ImageNet-1K, for all transformer layers. These plots are typical of the behavior we observed with all transformer backbones and architectures we considered. Clearly, $1 - \alpha$ starts with high to intermediate values, indicating that information flows through both branches of the mixer, but decays to $1 - \alpha \approx 0$ as training converges. Hence, as shown in Table 3, there is no degradation if the CCA branch is removed during inference. This eliminates a substantial amount of computation during inference, leading to the the training and inference setup of Figure 2, where CCA is not used at inference.

We explain this behavior by conjecturing that the downside of the computational efficiency of the BD-MLP is a more difficult learning problem, due to the independent processing of channel groups. This creates symmetries in the cost function, e.g. the order of the feature groups is not important, and requires a feature clustering operation that is likely to produce more local minima. The CCA branch helps to smooth out this cost function during training, while the feature groups are not established, by allowing inter-group communication. However, once the right feature groupings are found, CCA

---

[1]Since the features are normalized before the mixer, i.e centered such that $\mathbf{x}\,\mathbf{1}_N = 0$, $\mathbf{x}\mathbf{x}^T$ is the covariance matrix of features $\mathbf{x}$.

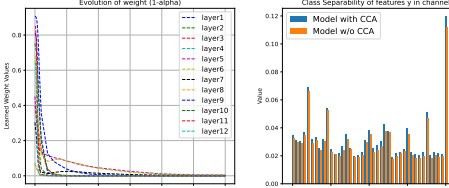 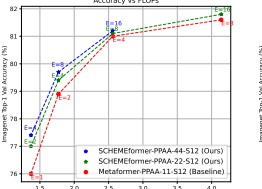 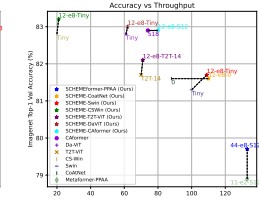

Figure 3: Impact of CCA (SCHEMEformer-44-e8-S12). **Left:** Evolution of weight $1 - \alpha$ across model layers. **Right:** Class separability of output features **y** (over 50 random classes of ImageNet-1K validation set) for model trained with and without CCA. See 6 for zoomed version.

Figure 4: SCHEME tradeoffs. **Left:** Accuracy vs FLOPs of various SCHEME models with different MLP configurations. **Right:** SCHEME mixer improves accuracy for fixed throughput or vice-versa for various popular ViT architectures. See 7 for zoomed version.

is no longer needed, and a simple BD-MLP mixer has no loss of performance over the standard MLP. Note that the operation of (5) is basically a projection of **x** into canonical subspaces of features that are correlated in the input image. This is likely to be informative to guide the group formation, but less useful when the features are already clustered. While this hypothesis is not trivial to test, since (5) varies from example to example, we confirmed that using CCA during training enhances class separability, which likely reduces overfitting for large expansion ratios. See the section 4.2 for more details.

**Computational Complexity:** The complexity of the BD-MLP block is controlled by the group numbers $g_1$, $g_2$ and the expansion factor $E$, with a total cost $\mathcal{O}(Ed^2/g_1 + Ed^2/g_2)$, where $d$ is the channel dimension. The computational cost of CCA is $\mathcal{O}(Nd^2)$ where $N$ is the number of tokens. Since CCA is not used during inference, it only adds to the computations during training. The SCHEME framework provides a systematic way to control the trade-off of transformer width vs depth, by controlling the block size and expansion hyperparameters.

### 3.2 THE SCHEMEFORMER FAMILY

The proposed SCHEME module enables efficient control of model complexity via the mixer hyperparameters $g_1$, $g_2$, and $E$. Table 1 shows that naively scaling down the ViT model by simply reducing $E$ causes a significant accuracy loss. The SCHEME module allows much more effective control of the accuracy/complexity trade-off, producing models of better performance for a fixed computational budget. This is demonstrated by the introduction of a new family of models, denoted as SCHEMEformer, obtained by replacing the channel mixer of the Metaformer-PPAA (52) architecture with the SCHEME module. Two such configurations are shown in Table 2 where the naming follows the convention {model-name}-{$g_1 g_2$}-e{$E$} where the model name is skipped for brevity.

## 4 EXPERIMENTAL RESULTS

### 4.1 COMPARISONS TO THE STATE OF THE ART

**Image Classification:** Image classification is evaluated on Imagenet-1K, without using extra data. We report the results with single crop top-1 accuracy at $224 \times 224$ input resolution. We evaluate the SCHEMEformer family of models based on the Metaformer-PPAA-S12 (52) obtained by replacing the MLP of the latter with the SCHEME module. Refer to Appendix for implementation details.

We start by evaluating how this improves the trade-off between model accuracy and complexity. As discussed in Section 3.1, when $g_1 = g_2$, a SCHEME transformer of expansion factor $gE$ has identical complexity to a standard transformer of expansion factor $E$. Hence, for fixed FLOPS, SCHEME allows an increase of the expansion factor by $g$. Figure 4 a) compares the performance of the Metaformer-PPAA-S12 with expansion ratios $E \in \{1, 2, 4, 8\}$ to comparable variants of the SCHEMEformer-PPAA-S12, with SCHEME mixers of either $g = 2$ (green curve) or $g = 4$ (blue curve) groups. The SCHEMEformer models have a better trade-off between accuracy and FLOPS, achieving higher accuracies for all complexity levels. Among these, the one with more feature groups ($g = 4$) has the best performance. While SCHEMEformer gains are observed for all FLOP levels, they are larger for lower complexity models. This makes SCHEME particularly attractive for the design of low complexity transformers, e.g. for edge devices or equivalent applications.

| Model | #Params (M) | FLOPs (G) | Thru (im/s) | Top-1 Acc (%) |
|---|---|---|---|---|
| gMLP-Ti (23) | 6 | 1.4 | - | 72.3 |
| ViT-L/16 (11) | 307 | 63.6 | 37 | 76.1 |
| Meta-11-e2-S12 (52) | 12 | 1.8 | 133 | 78.9 |
| MogaNet-T (20) | 5 | 1.10 | 44 | 79.0 |
| XCiT-T24 (12) | 12 | 2.3 | - | 79.4 |
| ViT-B/16 (11) | 86 | 17.6 | 112 | 79.7 |
| **SCHEME-44-S12** | 12 | 1.77 | **133** | **79.7** |
| $S^2$-MLP-deep (51) | 51 | 10.5 | - | 80.7 |
| Meta-S12 (52) | 17 | 2.6 | 87 | 81.0 |
| Swin-Tiny (25) | 29 | 4.5 | 100 | 81.3 |
| T2T-ViT t-14 (54) | 22 | 6.1 | 70 | 81.7 |
| DeiT-B (35) | 86 | 17.5 | 114 | 81.8 |
| ViL-Small (56) | 25 | 5.1 | - | 82.0 |
| **SCHEME-12-S12** | 21 | 3.35 | **130** | **82.0** |
| Focal-Tiny (50) | 29 | 4.9 | 29 | 82.2 |
| CPVT-Base (4) | 88 | 17.6 | - | 82.3 |
| DaViT-Tiny (8) | 23.0 | 4.3 | 61 | 82.8 |
| CSWin-Tiny (9) | 23.0 | 4.3 | 20 | 82.8 |
| **SCHEME-12-S24** | 40 | 6.47 | **69** | **82.8** |
| XCiT-L24 (12) | 189 | 36.1 | - | 82.9 |
| Swin-Small (25) | 50 | 8.7 | 31 | 83.0 |
| ViL-Base (56) | 56 | 13.4 | - | 83.2 |
| CSWin-Small (9) | 35 | 6.9 | 11 | 83.6 |
| **SCHEME-12-S36** | 58 | 9.58 | **38** | **84.0** |

Table 4: **Image Classification** on ImageNet-1K. Comparison with SOTA ViTs grouped by accuracy. Proposed SCHEME models use expansion ratio 8. SCHEMEformer family has higher throughput and accuracy than SOTA models.

| Backbone | #P (M) | F (G) | mIoU (%) |
|---|---|---|---|
| Semantic FPN | | | |
| ResNet-18(16) | 16 | 32.2 | 32.9 |
| PVT-Tiny(42) | 17 | 33.2 | 35.7 |
| ResNet-50(16) | 29 | 45.6 | 36.7 |
| PoolFormer-S12(52) | 16 | 30.9 | 37.2 |
| ResNet-101(16) | 48 | 65.1 | 38.8 |
| ResNeXt-101-32x4d(49) | 47 | 64.7 | 39.7 |
| PVT-Small(42) | 28 | 44.5 | 39.8 |
| XCiT-T12/8(12) | 8.4 | - | 39.9 |
| PoolFormer-S24(52) | 23 | 39.3 | 40.3 |
| SCHEMEformer-44-S12 | 15.5 | 34.3 | **40.9** |
| PVT-Medium(42) | 48 | 61.0 | 41.6 |
| PoolFormer-S36(52) | 35 | 47.5 | 42.0 |
| PVT-Large(42) | 65 | 79.6 | 42.1 |
| PoolFormer-M36(52) | 60 | 67.6 | 42.4 |
| SCHEMEformer-44-S24 | 24.8 | 45.7 | **42.5** |
| UperNet | | | |
| Swin-Tiny (25) | 60 | 945 | 44.5 |
| PVT-Large (43) | 65 | 318 | 44.8 |
| Focal-Tiny (50) | 62 | 998 | 45.8 |
| XCiT-S12/16 (12) | 52 | – | 45.9 |
| DaViT-Tiny (8) | 60 | 940 | 46.3 |
| SCHEME-DaViT-12-Tiny | 68 | 969 | **47.1** |

Table 5: **Semantic Segmentation** results on ADE20K. FLOPs calculated at $512 \times 512$ resolution for Semantic FPN and $1024 \times 1024$ input resolution for UperNet.

We next compare the SCHEMEformer family against the SOTA transformers in the literature. This is not an easy comparison, since models vary in size, FLOPS, and throughput. Because it is difficult to make any of these variables exactly the same for two different architectures, the comparison is only possible in terms of how the different architectures trade-off accuracy for any of the other factors. For a given pair of variables, e.g. FLOPS vs accuracy, the model is said to be on the Pareto frontier of the two variables if it achieves the best trade-off between the two. Table 4 presents a comparison of the SCHEMEformer family against various SOTA transformers in the literature. Each section of the table compares a SCHEMEformer model to a group of SOTA transformers of equivalent size or complexity. Note that, in each section, the remaining models have *both* lower throughput and accuracy than the SCHEMEformer model. In many cases they also have more parameters and FLOPs. Figure 1 provides a broader visualization of how SCHEME models establish Pareto frontiers for accuracy vs FLOPS, accuracy vs throughput, and accuracy vs model size (parameters). Figure 1 a) illustrates the trade-off between accuracy and FLOPS of many SOTA transformers. The dashed line connects the SCHEMEformer model results, summarizing the accuracy-FLOPs trade-off of the family. It can be seen that the SCHEME models lie on the Pareto frontier for these two objectives. This illustrates the fine control that SCHEME allows over the accuracy/complexity trade-off of transformer models. Figure 1 b) presents a similar comparison for model sizes. Like for FLOPS, the SCHEME models lie on the Pareto frontier for accuracy vs model size. In fact the two plots are quite similar, showing that in general there is a good correlation between model size and FLOPS.

This is not the case for throughput, which for transformers is known to not necessarily correlate with FLOPS, due to GPU parallelism. For example, Table 4 shows that, CSWin models have lower throughput despite having lower FLOPs while ViT has higher throughput despite having higher FLOPs. SCHEMEformer controls this trade-off by controlling the expansion ratio and block diagonal structure, which enables higher FLOPs utilization for a given throughput (14). We compute throughput on a NVIDIA-Titan-X GPU with a batch size of 1 with input size 224x224 averaged over 1000 runs. While comparisons could be made for larger batch sizes, we consider the setting for live/streaming applications, where speed is most critical. In these applications, the concern is usually inference throughput, which requires batch size of 1. Figure 1 c) illustrates the trade-off between accuracy and throughput of various models, including very fast ResNet models of low accuracy. It can bee seen that the SCHEMEformer again achieves the best trade-off between these two variables, thus lying on the Pareto frontier for accuracy vs throughput. Its performance is particularly dominant in the range of throughputs between 75 and 150 images/sec, where it significantly outperforms the other methods. These results demonstrate how the SCHEME module endows transformer designers

| RetinaNet 1× | #Par | T | $AP$ | $AP_{50}$ | $AP_{75}$ | $AP_S$ | $AP_M$ | $AP_L$ |
|---|---|---|---|---|---|---|---|---|
| PoolFormer-S12(52) | 21.7 | 13.1 | 36.2 | 56.2 | 38.2 | 20.8 | 39.1 | 48.0 |
| ResNet-50(16) | 37.7 | 15.7 | 36.3 | 55.3 | 38.6 | 19.3 | 40.0 | 48.8 |
| SCHEME-44-e8-S12 | 21 | 6.1 | **38.3** | **58.0** | **40.4** | **21.0** | **41.4** | **52.3** |
| PoolFormer-S24(52) | 31.1 | 8.9 | **38.9** | **59.7** | **41.3** | **23.3** | **42.1** | 51.8 |
| ResNet-101(16) | 56.7 | 12.1 | 38.5 | 57.8 | 41.2 | 21.4 | 42.6 | 51.1 |
| SCHEME-44-e8-S24 | 31 | 3.5 | 38.8 | 58.7 | 41.2 | 22.5 | 41.5 | **53.5** |
| DAT-T (47) | 38 | - | 42.8 | 64.4 | 45.2 | 28.0 | 45.8 | **57.8** |
| CrossFormer-S (44) | 41 | - | 44.4 | 55.3 | 38.6 | 19.3 | 40.0 | 48.8 |
| DaViT-Tiny (8) | 39 | 8.2 | 44.0 | 65.6 | 47.3 | 29.6 | 47.9 | 57.3 |
| SCHEME-DaViT | 47 | 7.8 | **44.7** | **66.2** | **48.3** | **30.0** | **48.8** | 57.2 |
| **Mask R-CNN 1×** | #Par | T | $AP^b$ | $AP_{50}^b$ | $AP_{75}^b$ | $AP^m$ | $AP_{50}^m$ | $AP_{75}^m$ |
| PoolFormer-S12(52) | 31.6 | 9.9 | 37.3 | 59.0 | 40.1 | 34.6 | 55.8 | 36.9 |
| ResNet-50(16) | 44.2 | 15.4 | 38.0 | 58.6 | 41.4 | 34.4 | 55.1 | 36.7 |
| SCHEME-44-e8-S12 | 31 | 6.0 | **39.8** | **61.9** | **42.9** | **37.1** | **59.2** | **39.4** |
| PoolFormer-S24(52) | 41.0 | 7.9 | 40.1 | 62.2 | 43.4 | 37.0 | 59.1 | 39.6 |
| ResNet-101(16) | 63.2 | 12.1 | 40.4 | 61.1 | 44.2 | 36.4 | 57.7 | 38.8 |
| SCHEME-44-e8-S24 | 41 | 3.4 | **40.9** | **62.5** | **44.6** | **37.8** | **59.7** | **40.4** |
| DAT-T (47) | 48 | - | 44.4 | 67.6 | 48.5 | 40.4 | 64.2 | 43.1 |
| CrossFormer-S (44) | 50 | - | 45.4 | 68.0 | 49.7 | 41.4 | 64.8 | **44.6** |
| DaViT-Tiny (8) | 48 | 7.8 | 45.0 | 68.1 | 49.4 | 41.1 | 64.9 | 44.2 |
| SCHEME-DaViT | 57 | 7.4 | **45.9** | **68.3** | **50.2** | **41.5** | **65.4** | 44.3 |

Table 6: **COCO-17 Object Detection and Instance Segmentation.** All backbones pretrained on ImageNet-1K (1x learning schedule). ($AP^b$, $AP^m$): (bounding box AP, mask AP). $T$: throughput (images/sec).

| Model | BDM | CCA | Acc (%) |
|---|---|---|---|
| Baseline | | | 78.9 |
| 44-e8-S12 | ✓ | | **79.1** |
| 44-e8-S12 | ✓ | ✓ | **79.7** |

Table 7: Contribution of BD-MLP and CCA branches in SCHEME.

| Module | #Par (M) | FLOPs (G) | T (img/s) | Top-1 Acc (%) |
|---|---|---|---|---|
| Shuffle | 11.83 | 1.77 | 108 | 79.1 |
| SE | 11.98 | 1.77 | 86 | 79.3 |
| Conv | 13.31 | 1.97 | 95 | 79.6 |
| DyCCA | 11.90 | 1.83 | 75 | 79.6 |
| **CCA** | **11.83** | **1.77** | **133** | **79.7** |

Table 8: Alternative feature attention designs in SCHEME.

| CCA | G1 (%) | G2 (%) | G3 (%) | G4 (%) | Ensemble (%) |
|---|---|---|---|---|---|
| | 51.1 | **54.3** | **55.4** | 54.7 | 73.2 |
| ✓ | 47.6 | 50.0 | 49.2 | **55.1** | **73.8** |

Table 9: **Ablation study** on the formation of feature clusters in the BD-MLP branch of the SCHEME module.

with the ability to produce multiple models at different points of the Pareto frontiers of accuracy vs model size, FLOPS, or throughput.

**SCHEME with other ViTs:** Figure 4 shows the Accuracy-Throughput curves for various SCHEME models obtained by replacing the MLP blocks of popular ViT architectures. It shows that SCHEME improves the accuracy for a fixed throughput, the throughput for a fixed accuracy, or both. The gains can be substantial, e.g. about $1\%$ accuracy gain (constant throughput) for the fastest transformer or speed gains of up to $20\%$ at constant accuracy. This shows that SCHEME benefits various ViT backbones, not just the MetaFormer.

**Semantic Segmentation:** Table 5 compares the semantic segmentation performance of two SCHEMEformer models using the semantic FPN framework (18) to various SOTA models of similar complexity, on ADE20K. Since we do not not have access to the throughput of most models, we report only parameter sizes and FLOPS. In each section of the table, the remaining models have comparable or larger FLOPs and model size but lower accuracy than the corresponding SCHEME-former. For example, SCHEMEformer-44-e8-S12 achieves 40.9% mIoU, which is 5.2/3.7 points higher than the PvT-Tiny/PoolFormer-S12, which both have comparable size and FLOPs. Similarly, the S24 model outperforms PoolFormer-M36 using only 41% of its parameters. To demonstrate the applicability of SCHEME to larger models, we also present a comparison of the SCHEME version of the DaViT-Tiny using the UperNet framework (48). While the DaViT-Tiny is already the best model in the table, the use of the SCHEME mixer improves its performance by an additional **0.8** points.

**Object Detection:** Table 6 compares SCHEMEformer models to models of similar complexity on the COCO-17 object detection and instance segmentation benchmark, for both Retinanet and Mask-RCNN detection heads. Again, the SCHEMEformer models outperform most other models of the same or smaller size. The only exception is the PoolFormer-S24, which slightly outperforms (0.1 points) the comparable SCHEME-former-44-S24, for the RetinaNet head. However, with the stronger Mask R-CNN head, the SCHEMEformer-44-S24 beats the PoolFormer-S24 by 0.8 points. For the top performing models in the bottom third of each section of Table 6, SCHEME-DaViT-Tiny improves over DaViT-Tiny by an additional **0.7%** and **0.9%**, for RetinaNet and MaskRCNN heads respectively, while maintaining a comparable throughput.

## 4.2 ABLATION STUDIES

**Contribution of BD-MLP and CCA branch:** Table 7 shows an ablation of the contribution of the BD-MLP and CCA branches. Starting from the Metaformer-PPAA-11-e2-S12, with expansion ratio $E = 2$ and dense MLP, we replace the channel mixer by SCHEME to obtain the SCHEMEformer-PPAA-44-e8-S12 ($g_1 = g_2 = 4$ and $E = 8$), which maintains the number of parameters and FLOPs

constant. The SCHEME model with only BD-MLP improves on the baseline by **0.2**%. The addition of the CCA branch provides an additional gain of **0.6**%, showing the gains of better feature clusters. Since CCA is not used at inference, its gains are *free* in terms of additional parameters/FLOPs.

**Regularizing effect of CCA:** Fig. 3 shows the evolution of the weight $1 - \alpha$ of the CCA branch in (6) during training. While initially large, it gradually decays to zero as training progresses. This holds for all network layers. Hence, CCA can be discarded at inference. Fig. 1 in supplementary plots the weights $1 - \alpha$ upon training convergence, for the family of SCHEMEformer-PPAA-44-e8 models, confirming that the weights are indeed very close to zero across all layers.

**Effect of large expansion ratios:** The left of Figure 4 shows the effect of simultaneously increasing the expansion ratio $E$ and adjusting groups to realize different models of similar size and complexity. All models are based on the Metaformer-PPAA-S12. For fixed parameters/FLOPs, SCHEMEformer models achieve a gain of **1.0**% to **1.4**% over the baseline by increasing $E$ from 1 to 4. The gains increase with larger expansions and saturate at larger FLOPs. This shows that higher internal feature dimensions are important for obtaining better accuracy with smaller ViT models.

**Alternative designs of Channel Mixer:** We investigate whether alternative choices to the CCA branch could accomplish this goal more effectively. Table 8 compares models that replace CCA with other feature mixing operations: the channel shuffling operation of ShuffleNet (57), a squeeze and excitation (17) network (SENet), a single layer of convolution, and a dynamic version of CCA (DyCCA), where the weight $\alpha$ of (6) is predicted dynamically, using GCT attention (28). CCA obtains the best result. While CCA is computationally heavier than some of these alternatives, it is not needed at inference, as shown in Table 3. This is not true for the alternatives, which produce much more balanced weights $\alpha$ after training convergence, and cannot be discarded at inference without performance drop. We conjecture that, because the alternatives have learnable parameters, the network learns to use them to extract complementary features, which must be used at inference.

**Feature Clustering:** We conjectured above that CCA helps training because it facilitates feature clustering into naturally independent groups that do not require inter-group communication. We tested this hypothesis by studying the intermediate feature vectors $\mathbf{v}_l$ (obtained after max-pooling the features $\mathbf{y}$ of (4)) extracted from four randomly selected layers $l$ of two ImageNet pretrained models, trained with and without CCA. We split the features into the 4 groups used in the model $\mathbf{v}_{l,g}, g \in \{1, \dots 4\}$ and concatenated the features of all layers in the same group. This produced four vectors $\mathbf{u}_g = \text{concat}(\{v_{l,g}\}_l)$ containing the features of each group $g$ extracted throughout the network. A linear classifier was then learned over each vector $\mathbf{u}_g$. Table 9 shows the top-1 accuracy per feature group and model. To evaluate whether groups learn different class clusters, we also average the outputs from the four group classifiers to obtain the final accuracy. Without CCA, i.e. no group communication during training, the network produces feature groups individually more predictive of the image class, but less predictive when combined. This suggests that there is redundancy between the features of the different groups. By introducing inter-group communication, CCA enables the groups to learn more diverse features, that complement each other. Fig. 3 shows the class separability of the intermediate features $\mathbf{y}$ of (4) of a randomly chosen layer of the SCHEMEformer-PPAA-44-e8-S12. Class separability was measured as in (30), with a final value obtained by averaging the class separability across all classes. The model trained with CCA has higher class separability than that without it. This confirms that CCA is helpful in forming feature clusters that increase class separability during training. Conversely, we tested if CCA is helpful when channel shuffling is inserted in between the two mixer MLP layers, which destroys the group structure. This variant of the SCHEMEformer-PPAA-44-e8-S12 model achieved an accuracy of 79.1% for both training with and without CCA (Table 7, row 2). This shows that CCA is not helpful when feature groups are mixed. Similarly, CCA did not provide any gains when applied to the standard MLP branch with full feature mixing. These results suggest that CCA indeed helps to form the independent feature clusters needed to achieve the computational efficiency of channel groups without performance degradation. Table 9 shows that, despite the higher accuracy of the individual group features of the model trained without CCA, the model trained with CCA has **0.6**% higher ensemble accuracy.

## 5 LIMITATION AND CONCLUSION

Although SCHEME module improves the accuracy-throughput curves of popular ViTs, it incurs a slight overhead in memory (see appendix section A.3.2) during training due to the channel attention operation which limits the batch size on smaller GPUs. However, Figure 3 shows that the contribution of the CCA branch becomes negligible after 150 epochs. Therefore, the CCA branch can be removed beyond this point, significantly reducing the training overhead.

In this work, we proposed the SCHEME module for improving the performance of ViTs. SCHEME leverages a block diagonal feature mixing structure to enable MLPs with larger expansion ratios, a property that is shown to improve transformer performance, without increase of model parameters or computation. It uses a weighted fusion of a BD-MLP branch, which abstracts existing MLPs with block diagonal structure, and a parameter-free CCA branch that helps to cluster features into groups during training. The CCA branch was shown to improve training but not be needed at inference. Experiments showed that it indeed improves the class separability of the internal feature representation of the BD-MLP branch, helping create feature clusters that are informative of the image class. The standard transformer MLP was replaced with the SCHEME module to obtain a new family of SCHEMEformer models with improved performance for classification, detection, and segmentation, for fixed parameters and FLOPs with favorable latency. SCHEME was shown to be effective for various ViT architectures and to provide a flexible way to scale models, always outperforming models with smaller MLP expansion ratios having the same complexity.

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

# A APPENDIX

## A.1 CODE RELEASE

Code and trained models will be released upon acceptance of the paper.

## A.2 IMPLEMENTATION DETAILS

### A.2.1 HYPERPARAMETER SETTINGS

Table 11 shows the detailed hyperparameter settings of the family of SCHEMEformer models reported in the main paper.

## A.3 ABLATION STUDIES

### A.3.1 PLOT OF THE LEARNED $1 - \alpha$ WEIGHTS IN THE SCHEME MODULE FOR SCHEMEFORMER MODELS

The learned weights $1 - \alpha$ for the SCHEMEformer model family are shown in Fig. 8. Interestingly, the learned weights coarsely approximate the shape of a gaussian distribution. The learned weights reach a peak value in the middle layers of the network and drop to zero for all the other layers. The middle of the network typically corresponds to the initial few layers of the third stage of the model that contains the maximum number of transformer blocks for all the models shown in Fig. 8. We conjecture that the weights for these layers have not fully converged and that feature mixing can still be useful for these layers and so training for more epochs will allow the $1 - \alpha$ weights of these layers to converge to zero. To test this hypothesis, we trained the SCHEMEformer-PPAA-12-e8-S12 model for an additional 200 epochs beyond the standard 300 epochs and observed that the peak value of the $1 - \alpha$ weights decreased further by 0.11 as compared to the model trained for 300 epochs and the accuracy improved by **0.4**%. The weight norm (of all layers) decreased from 0.18 for 300 epoch model to 0.05 for 500 epoch model. This confirms our hypothesis and training SCHEMEformer models for larger epochs can further improve the accuracy.

### A.3.2 TRAINING OVERHEAD OF CCA

Table 12 compares the training GPU memory and throughput for SCHEME mixer with and without using CCA. CCA improves the accuracy by 0.6% with only a slight increase in the GPU memory (+12.5%) and training time (+16.7%). Further, CCA is not needed during inference thereby providing gains for "free" without additional computational cost at inference.

### A.3.3 TRAINING FOR LONGER EPOCHS

Table 13 shows the comparison of training for longer epochs for SCHEMEformer with the baseline Metaformer model. We train for an additional 200 epochs from the standard 300 epochs. SCHEMEformer-PPAA-44-e8-S12 trained for 300 epochs even outperforms the baseline model trained for 500 epochs. On continuing the training from 300 to 500 epochs, SCHEMEformer continues to improve the performance without saturation suggesting that it is beneficial to train with CCA for longer epochs. 300 to 500 epochs is a much larger increase of training time (67%) than the 16.7% increase in training time required by SCHEME mixer (see Table 12).

### A.3.4 IMPACT OF REMOVING CCA AT INFERENCE

Table 14 shows the impact of removing CCA at inference, for various backbones. While the number of FLOPS decreases, the top-1 accuracy changes very little ($\approx 0.02$ difference). Hence, there is no advantage in using CCA at inference. This is unlike training, where the use of CCA makes a non-negligible difference, as shown in Table 6 of the main paper.

Table 15 and 16 show the results of removing CCA at inference for object detection and semantic segmentation models, respectively. For both tasks, the results are identical to the model using CCA showing that the CCA also generalizes to downstream tasks.

### A.3.5 CCA

In the main paper, feature groups across different layers of the model was used to demonstrate the learning of feature clusters by CCA. Here, the feature from the final layer of the model is only considered. Table 18 shows the top-1 accuracy per feature group and model. The average of the outputs from the four group classifiers is reported in the final column of the table. The effect is more pronounced when using a single layer feature with **+1.09**% accuracy difference between the model with and without CCA. This further reinforces that by introducing inter-group communication, CCA enables the groups to learn more diverse sets of features, that complement each other.

### A.3.6 LARGER EXPANSION RATIOS

Table 17 shows additional results of using larger expansion ratios with SCHEME mixer (illustrated in Fig. 3 of the main paper) for the same number of parameters and FLOPs using the Metaformer-PPAA-S12 baseline model. We observe that SCHEMEformer-PPAA consistently outperforms the baseline for larger expansions ratios with larger gains at lower FLOPs. The performance saturates as the model size and FLOPs increases.

## A.4 QUALITATIVE ANALYSIS

### A.4.1 GRAD-CAM (31) VISUALIZATION

Fig. 9 shows the results of class activation maps for SCHEMEformer-PPAA-44-e8-S12 model for a few examples from the validation set of ImageNet-1K dataset. The stronger heatmap responses around the salient features of an object (e.g., body of a bird, cat) shows that the model ignores the background and attends to more discriminative spatial regions. Fig. 9 also shows the qualitative comparison with a few existing methods such as ResNet-50, DeiT-S, Poolformer etc. of similar complexity. It demonstrates that SCHEMEformer-PPAA-44-e8-S12 attends to the complete object class and less spurious features showing that it is better than the competing methods.

### A.4.2 ATTENTION VISUALIZATION

Figure 10 shows the results of attention maps with and without CCA on Imagenet dataset. We find that there are differences in attention between using or not using CCA. CCA increases the tendency of the model to attend to the regions where the class is present, which is denoted by green ellipses.

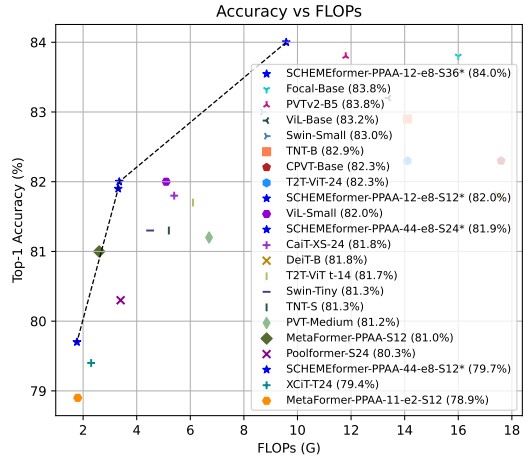

(a) Accuracy-FLOPs

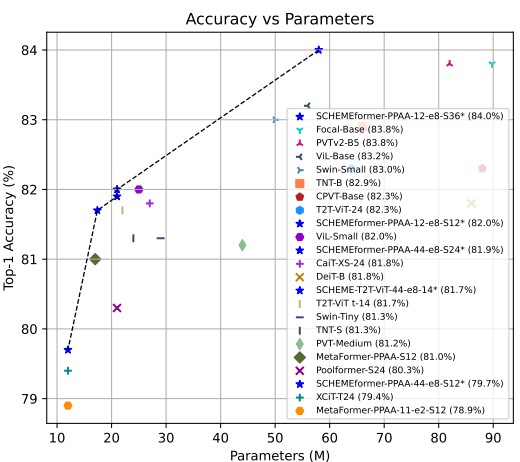

(b) Accuracy-Parameters

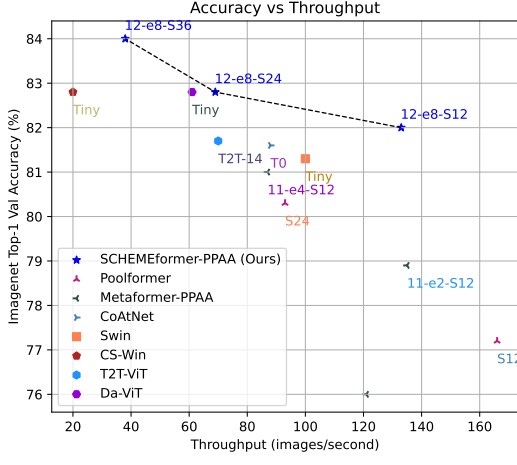

(c) Accuracy-Throughput

Figure 5: [Zoomed version] Comparison of the proposed SCHEMEformer family, derived from the Metaformer-PPAA-S12 model (52) with higher expansion ratios in the MLP blocks, and many SOTA transformers from the literature. The SCHEMEFormer family establishes a new Pareto frontier (optimal trade-off) for a) accuracy vs. FLOPs, b) accuracy vs model size, and c) accuracy vs, throughput. SCHEMEformer models are particularly effective for the design of fast transformers (throughput between 75 and 150 images/s) with small model size.

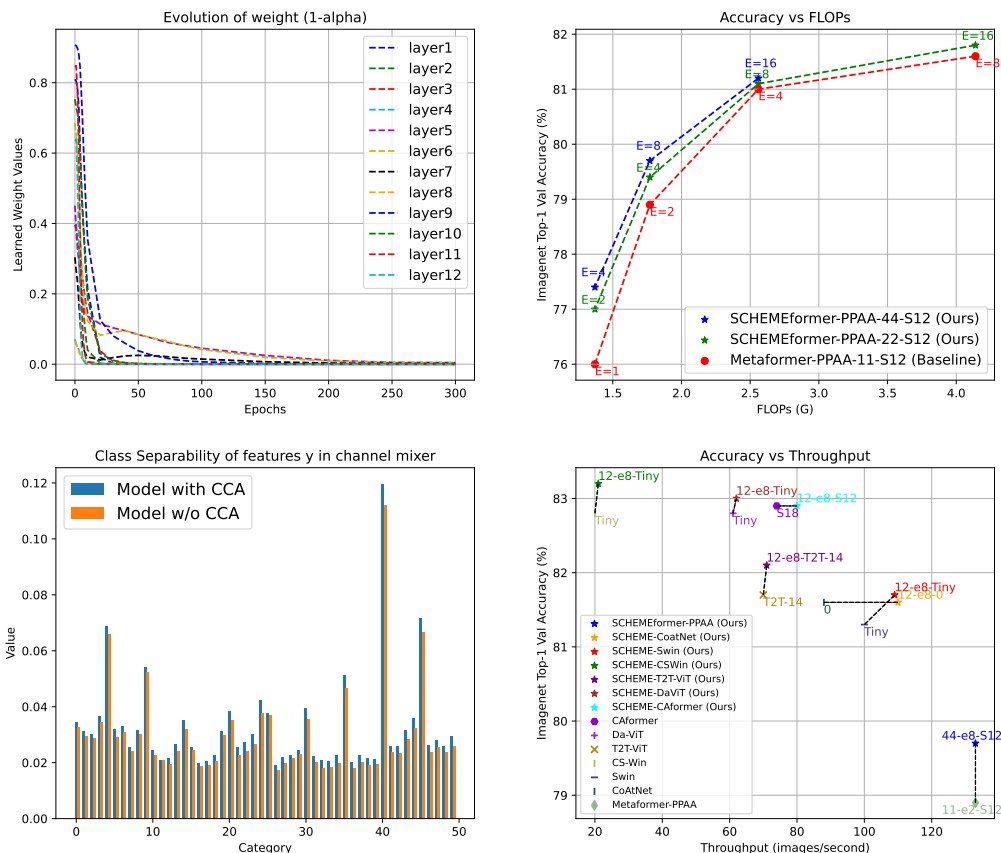

Figure 6: [Zoomed version] Impact of CCA (SCHEMEformer-44-e8-S12). **Top:** Evolution of weight $1-\alpha$ across model layers. **Bottom:** Class separability of output features $\mathbf{y}$ (over 50 random classes of ImageNet-1K validation set) for model trained with and without CCA.

Figure 7: [Zoomed version] SCHEME tradeoffs. **Top:** Accuracy vs FLOPs of various SCHEME models with different MLP configurations. **Bottom:** SCHEME mixer improves accuracy for fixed throughput or vice-versa for various popular ViT architectures.

| Model | #Par (M) | FLOPs (G) | T (img/s) | Acc (%) |
|---|---|---|---|---|
| ViT-Base (11) | 86 | 17.6 | 112 | 79.7 |
| SCHEME-ViT-12-e8-Base | 77.1 | 15.5 | **130** | **79.9** |
| DeiT-Tiny (35) | 6 | 1.3 | 117 | 74.5 |
| SCHEME-DeiT-12-e8-Tiny | 7.5 | 1.6 | **117** | **76.0** |
| Poolformer-S12 | 12.0 | 1.8 | 166 | 77.2 |
| SCHEME-Poolformer-12-e8-S12 | 16.7 | 2.6 | **171** | **78.5** |
| Metaformer-S12 | 12.0 | 1.8 | 133 | 78.9 |
| SCHEMEformer-PPAA-12-e8-S12 | 11.8 | 1.8 | **133** | **79.7** |
| CAformer-S12 | 25 | 4.20 | 74 | 82.9 |
| SCHEME-CAformer-12-e8-S12 | 23.9 | 3.60 | **80** | **82.9** |
| CoAtNet-0 | 25.0 | 4.2 | 88 | 81.6 |
| SCHEME-CoAtNet-12-e8-0 | 24.0 | 4.1 | **110** | **81.6** |
| CSWin-Tiny | 23.0 | 4.3 | 20 | 82.8 |
| SCHEME-CSWin-12-e8-Tiny | 29.1 | 5.6 | **21** | **83.2** |
| DaViT-Tiny | 23.0 | 4.3 | 61 | 82.8 |
| SCHEME-DaViT-12-e8-Tiny | 37.0 | 6.6 | **62** | **83.0** |
| T2T-ViT-14 | 21.5 | 6.1 | 70 | 81.7 |
| SCHEME-T2T-ViT-12-e8-14 | 27.7 | 8.1 | **71** | **82.1** |
| BiFormer-Tiny(58) | 13 | 2.2 | 57 | 81.4 |
| SCHEME-BiFormer-12-e3-Tiny | 11.5 | 1.9 | **59** | **81.4** |

Table 10: Comparison with state-of-the-art ViT models on the ImageNet-1K dataset. The SCHEME module enhances the accuracy of existing ViTs while maintaining or achieving higher throughput.

Table 11: Hyperparameter Settings for the family of SCHEMEformer models trained on ImageNet-1K dataset.

| Model | SCHEMEformer-PPAA-44-e8 | | | SCHEMEformer-PPAA-12-e8 | | | SCHEME-CAformer | |
|---|---|---|---|---|---|---|---|---|
| | S12 | S24 | S36 | S12 | S24 | S36 | 44-e8-S18 | 12-e8-S12 |
| Peak drop rate of stoch. depth $d_r$ | 0.1 | 0.2 | 0.4 | 0.1 | 0.2 | 0.4 | 0.15 | 0.15 |
| LayerScale initialization $\epsilon$ | $10^{-5}$ | $10^{-5}$ | $10^{-6}$ | $10^{-5}$ | $10^{-5}$ | $10^{-6}$ | $10^{-5}$ | $10^{-5}$ |
| Data augmentation | AutoAugment | | | | | | | |
| Repeated Augmentation | off | | | | | | | |
| Input resolution | 224 | | | | | | | |
| Epochs | 300 | | | | | | | |
| Hidden dropout | 0 | | | | | | | |
| GELU dropout | 0 | | | | | | | |
| Classification dropout | 0 | | | | | | | |
| Random erasing prob | 0.25 | | | | | | | |
| EMA decay | 0 | | | | | | | |
| Cutmix $\alpha$ | 1.0 | | | | | | | |
| Mixup $\alpha$ | 0.8 | | | | | | | |
| Cutmix-Mixup switch prob | 0.5 | | | | | | | |
| Label smoothing | 0.1 | | | | | | | |
| Batch size used in the paper | 1024 | | | | | | | |
| Learning rate decay | cosine | | | | | | | |
| Weight decay | 0.05 | | | | | | | |
| Gradient clipping | None | | | | | | | |
| Warmup epochs | 5 | | | | | | 20 | |
| Relation between peak learning rate and batch size | $lr = \frac{batch\ size}{1024} \times e^{-3}$ | | | | | | $lr = \frac{batch\ size}{1024} \times 8 \times e^{-3}$ | |
| Optimizer | AdamW | | | | | | LAMB | |
| Adam $\epsilon$ | $1e^{-8}$ | | | | | | None | |
| Adam $(\beta_1, \beta_2)$ | (0.9, 0.999) | | | | | | None | |

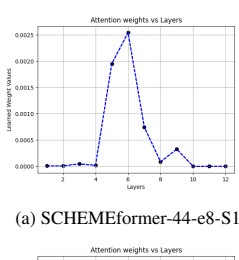 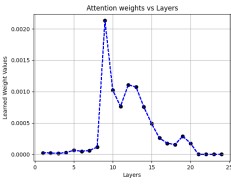 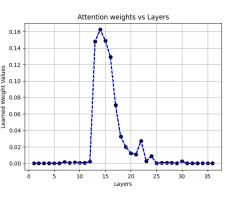 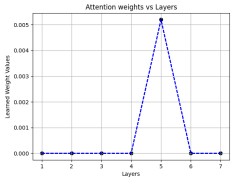

(a) SCHEMEformer-44-e8-S12  (b) SCHEMEformer-44-e8-S24  (c) SCHEMEformer-44-e8-S36  (d)  SCHEME-CoAtNet-44-e8-Tiny

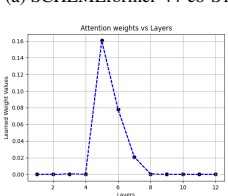 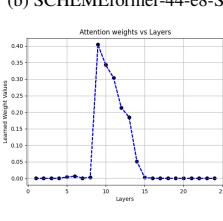 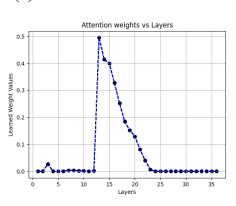

(e) SCHEMEformer-12-e8-S12  (f) SCHEMEformer-12-e8-S24  (g) SCHEMEformer-12-e8-S36  (h) SCHEME-Swin-12-e8-Tiny

Figure 8:  Plot of the learned weight (1-$\alpha$) values across different layers of a network for the family of SCHEMEformer models.  The weights reach a peak value near the middle of the network.  We demonstrate that these peak weights are not yet converged and training the network for more epochs decays these weights to zero while also improving the accuracy.  For example, training the SCHEMEformer-PPAA-12-e8-S12 model for 200 additional epochs reduced the weight norm of the vector of $1 - \alpha$ weights from 0.18 to 0.05 showing that these weights gradually approach zero as the training progresses while improving the accuracy further by 0.4%.

Table 12: **Training Overhead of CCA.** CCA adds only a small overhead in GPU memory and training time.

| Model | CCA | #Par (M)↓ | Train FLOPs (G)↓ | Val Acc (%) ↑ | GPU Mem. (G) ↓ | Train Throughput (iters/s) ↑ |
|---|---|---|---|---|---|---|
| SCHEMEformer-PPAA-44-e8-S12 | | **11.8** | **1.77** | 79.1 | **8** | **215** |
| SCHEMEformer-PPAA-44-e8-S12 | ✓ | **11.8** | 2.16 | **79.7** | 9 | 180 |

Table 13: **Ablation study** of longer training for SCHEMEformer-PPAA-44-e8-S12.

| Model | #P (M) | FLOPs (G) | Throughput (img/s) | 300-Acc (%) | 500-Acc (%) |
|---|---|---|---|---|---|
| Metaformer-11-e2-S12 (Baseline) | 11.8 | 1.77 | 133 | 78.9 | 79.6 |
| SCHEMEformer-44-e8-S12 | 11.8 | 1.77 | 133 | **79.7** | **80.1** |

Table 14: **Impact of removing CCA branch** during inference.

| Model | CCA Used | #Params (M) | FLOPs (G) | Top-1 Acc (%) |
|---|---|---|---|---|
| SCHEMEformer-PPAA-44-e8-S12 | ✓ | 11.83 | 2.16 | 79.74 |
| SCHEMEformer-PPAA-44-e8-S12 | | 11.83 | 1.77 | 79.72 |
| SCHEMEformer-PPAA-12-e8-S24 | ✓ | 40.0 | 7.3 | 82.80 |
| SCHEMEformer-PPAA-12-e8-S24 | | 40.0 | 6.5 | 82.76 |
| SCHEMEformer-PPAA-12-e8-S36 | ✓ | 58.8 | 10.8 | 84.00 |
| SCHEMEformer-PPAA-12-e8-S36 | | 58.8 | 9.6 | 83.95 |
| SCHEME-CoatNet-44-e8-0 | ✓ | 17.80 | 3.83 | 80.70 |
| SCHEME-CoatNet-44-e8-0 | | 17.80 | 3.42 | 80.68 |
| SCHEME-Swin-12-e8-T | ✓ | 36.93 | 7.00 | 81.69 |
| SCHEME-Swin-12-e8-T | | 36.93 | 5.89 | 81.69 |

Table 15: **Ablation study of removing CCA for COCO-17 Object Detection and Instance Segmentation.** Removing CCA at inference does not impact the AP values as they are identical to the model using CCA at inference. $AP^b$ and $AP^m$ denote bounding box AP and mask AP, respectively. Backbone models denote SCHEMEformer-PPAA-44-e8- variants.

| Backbone | CCA Used | #P | $AP$ | $AP_{50}^b$ | $AP_{75}^b$ | $AP_S^b$ | $AP_M$ | $AP_L$ | #P | $AP^b$ | $AP_{50}^b$ | $AP_{75}^b$ | $AP^m$ | $AP_{50}^m$ | $AP_{75}^m$ |
|---|---|---|---|---|---|---|---|---|---|---|---|---|---|---|---|
| | | | | RetinaNet 1× | | | | | | | Mask R-CNN 1× | | | | |
| S12 | ✓ | 21 | 38.3 | 58.0 | 40.4 | 21.0 | 41.4 | 52.3 | 31 | 39.8 | 61.9 | 42.9 | 24.1 | 53.0 | 42.3 |
| S12 | | 21 | 38.3 | 58.0 | 40.4 | 21.0 | 41.4 | 52.3 | 31 | 39.8 | 61.9 | 42.9 | 24.1 | 53.0 | 42.3 |
| S24 | ✓ | 31 | 38.8 | 58.7 | 41.2 | 22.5 | 41.5 | 53.5 | 41 | 40.9 | 62.5 | 44.6 | 24.6 | 55.6 | 43.8 |
| S24 | | 31 | 38.8 | 58.7 | 41.2 | 22.5 | 41.5 | 53.5 | 41 | 40.9 | 62.5 | 44.6 | 24.6 | 55.6 | 43.8 |

Table 16: **Ablation study** of removing CCA for Semantic Segmentation results on ADE20K validation dataset.

| | CCA | Semantic FPN | | |
|---|---|---|---|---|
| | | #Par | FLOPs | mIoU (%) |
| | ✓ | 15.5 | 36.4 | 40.9 |
| | | 15.5 | 34.3 | 40.9 |
| | ✓ | 24.8 | 49.8 | 42.5 |
| | | 24.8 | 45.7 | 42.5 |

Table 17: **Ablation study** on the effect of larger expansion ratios in BD-MLP block of SCHEME on ImageNet-1K validation dataset.

| Model | #Par (M) | FLOPs (G) | Top-1 Acc (%) |
|---|---|---|---|
| Metaformer-PPAA-11-e1-S12 | 9.6 | 1.37 | 76.0 |
| **SCHEMEformer-PPAA-22-e2-S12** | 9.6 | 1.37 | **77.0** |
| **SCHEMEformer-PPAA-44-e4-S12** | 9.6 | 1.37 | **77.4** |
| **SCHEMEformer-PPAA-66-e6-S12** | 10.0 | 1.45 | **77.9** |
| Metaformer-PPAA-11-e2-S12 | 11.8 | 1.77 | 78.9 |
| **SCHEMEformer-PPAA-22-e4-S12** | 11.8 | 1.77 | **79.4** |
| **SCHEMEformer-PPAA-44-e8-S12** | 11.8 | 1.77 | **79.7** |
| **SCHEMEformer-PPAA-33-e6-S12** | 12.5 | 1.87 | **79.8** |
| **SCHEMEformer-PPAA-22-e6-S12** | 14.1 | 2.16 | **80.4** |
| Metaformer-PPAA-11-e4-S12 | 16.5 | 2.56 | 81.0 |
| **SCHEMEformer-PPAA-22-e8-S12** | 16.5 | 2.56 | **81.1** |
| **SCHEMEformer-PPAA-44-e16-S12** | 16.5 | 2.56 | **81.2** |

Table 18: **Ablation study** on the formation of feature clusters in the BD-MLP branch of the SCHEME module. We train a linear classifier on top of the four feature groups extracted from the final MLP mixer of the transformer block of the network. The model trained with CCA forms feature clusters that learn diverse and complementary set of features that can obtain 1% higher validation accuracy than the model trained without CCA.

| Model | CCA | Group1 (%) | Group2 (%) | Group3 (%) | Group4 (%) | Ensemble (%) |
|---|---|---|---|---|---|---|
| SCHEMEformer-44-e8-S12 | | **29.63** | 37.59 | **37.57** | 35.89 | 73.95 |
| SCHEMEformer-44-e8-S12 | ✓ | 27.89 | **37.64** | 30.02 | **37.67** | **75.04** |

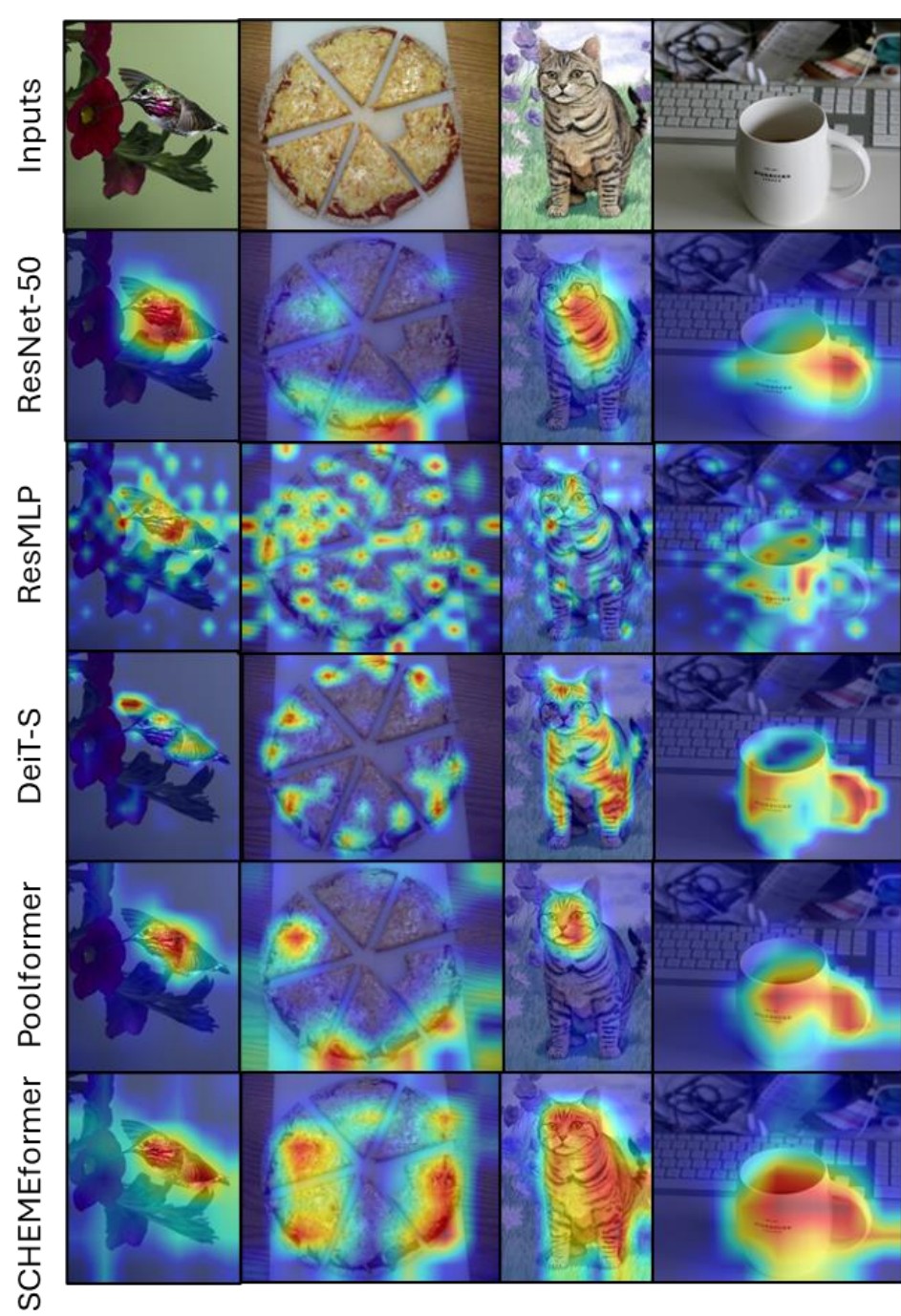

Figure 9: GRAD-CAM (31) visualization for a few validation samples on ImageNet-1K dataset for SCHEMEformer-PPAA-44-e8-S12 model and comparison with other competing methods.

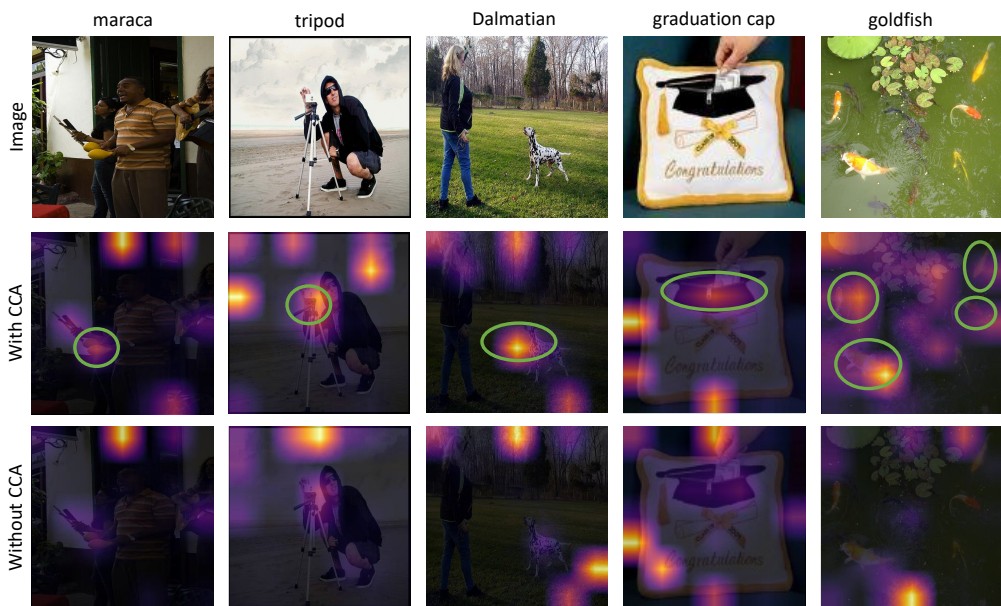

Figure 10: Visualization of last layer attention maps of the SCHEMEformer-PPAA-44-e8-S12 model with and without CCA on Imagenet validation dataset images. The ellipses identify the region of the object in the target class. Since the SCHEMEformer model does not have class tokens, the attention maps are not necessarily interpretable. The spikes in the attention maps are also common in attention visualizations and can be avoided by using registers (7).

