# OpenReview forum: "SCHEME: Scalable Channel Mixer for Vision Transformers"
_ICLR.cc/2025/Conference — ICLR 2025 Conference Withdrawn Submission_

### Official Review · Reviewer_JU1Q · 2024-10-29

**Soundness:** 4
**Presentation:** 3
**Contribution:** 4
**Rating:** 6
**Confidence:** 4

**Summary:**

This paper concentrate on the  channel mixer or feature mixing block design and propose a block diagonal MLP structure with a lightweight, parameter-free, channel covariance attention (CCA) module to improve the feature clusters formed. Replacing the vanilla MLP with proposed SCHEME module, various experiments were conducted to validate its performance.

**Strengths:**

1. Channel mixer design, the focus of this paper, is a more than interesting topic for the backbone design.
2. Comprehensive vision experiments (*e.g.*, classification, detection, segmentation) and analysis (*e.g.* `Paragraph 3`in `Sec 4.2` for channel covariance attention) were conducted to validate the proposed method's performance.

**Weaknesses:**

1. With regard to the design of channel mixer, there exists many previous research, such as gMLP[1], channel aggregation module in moganet[2], S2-MLP[3]. In addition to comparing with models using traditional MLP, these should also be taken into account.
2. More scaling experiments are needed. In this paper, the biggest model was 58M in `Tab.1`, which may could only match the `base-size` model in general. To comprehensively validate the method's scaling capability in backbone, `Large` or even `X-Large` model are also need.
3. Typos for the dimension of $W_1$ in `Line 196,221`, $N$ -> $d$.

I would be more than willing to reconsider the rating based on the authors' response.

Refs:

[1] Liu, Hanxiao et al. “Pay Attention to MLPs.” Neural Information Processing Systems (2021).

[2] Li, Siyuan et al. “MogaNet: Multi-order Gated Aggregation Network.” International Conference on Learning Representations (2022).

[3] Yu, Tan et al. “S2-MLP: Spatial-Shift MLP Architecture for Vision.” 2022 IEEE/CVF Winter Conference on Applications of Computer Vision (WACV) (2021): 3615-3624.

**Questions:**

See `Weaknesses`.

---

> ### Author Response · Authors · 2024-11-26
>
> We thank the reviewer for their comments and useful feedback.
>
>
> **$W1$**:  We have included comparisons with the methods mentioned by the reviewer in Table 4. We observe that the MLP mixer methods such as gMLP and S2-MLP generally exhibit lower accuracies compared to their attention-based counterparts as typically noted in the ViT literature. While MogaNet demonstrates improved accuracy, it is significantly slower than SCHEME and other similarly accurate variants due to the additional computational overhead from multiple split and depthwise convolution operations. We also wish to clarify that since SCHEME is a generic framework, it can be readily integrated into architectures like gMLP or S2-MLP.
>
>
> **$W2$**:  Due to computational resource constraints, we are unable to scale up models beyond 60 M parameters and so we restricted our focus on models below this threshold. In the revised version, we have included additional comparisons in Table 10, incorporating more recent variants, bringing the total to 10 different ViT model evaluations. We believe this demonstrates the generalizability and scalability of SCHEME, which can be readily extended to models exceeding 60M parameters.
>
> **$W3$**:  Thank you for pointing out the typo. We fixed it in the revision.

---

> > ### Comment · Reviewer_JU1Q · 2024-11-27
> > **Response To Authors' Rebuttal.**
> >
> > Thanks for authors' rebuttal, which has addressed some my previous concern about this paper. After reading the revised version, I decide to revise my rating to 6.

---

### Official Review · Reviewer_vWvm · 2024-10-31

**Soundness:** 2
**Presentation:** 2
**Contribution:** 2
**Rating:** 5
**Confidence:** 4

**Summary:**

This paper studies the channel-mixer of modern vision architectures, and propose the SCHEME module, a new channel-mixer that split features to multiple sub channel groups and project these channel groups to lager dimensionality for learning richer internal representation. A channel covariance attention is designed to make information exchange among sub channel groups during training time. The experiments show that the proposed SCHEME outperforms the traditional channel-mixers (e.g., FFN/MLP) in several tasks.

**Strengths:**

1. The paper is well-structured, providing a clear and accessible overview of the motivation and methodology behind the proposed approach. The logical flow enables readers to easily understand both the problem addressed and the design of the proposed solution.

2. The methodology section is well-developed, with good supporting arguments for each component. The progression from MLP to BD-MLP, followed by the integration of CCA, is clear, well-motivated.

3. The use of CCA as a regularizer for BD-MLP to facilitate interaction across channel groups during training, while discarding it during inference to streamline computational efficiency, is an innovative and effective approach.

4. The proposed SCHEME module and its various configurations demonstrate impressive performance across a range of tasks.

**Weaknesses:**

1. The labels and coordinates in Figures 1, 3, and 4 are too small, making them challenging to interpret. For a basic academic paper, readability of figure details is essential.

2. The effectiveness of the proposed SCHEME is primarily validated on MetaFormer. While the authors briefly present results on other backbones like Swin and DaViT in Figure 4 (yet difficult to interpret). SCHEME’s generalizability is limited by testing primarily on ViT-based architectures, lacking ablation on modern ConvNet architectures (e.g., ConvNeXt [1], FasterNet [2], and InceptionNeXt [3]) that also use FFN/MLP modules.

3. Although SCHEME shows theoretical complexity advantages, its on-device efficiency is underexplored. Only limited throughput comparisons are presented in Tables 4 and 8. Since an on-device benchmark has been implemented, including detailed on-device results across ablation and comparison studies would substantiate claims about SCHEME’s efficiency over conventional FFNs. Additionally, details on the benchmark configurations, hardware, and input shapes are necessary to ensure fair comparisons.

4. The CCA component functions as a channel mixer and resembles linear attention mechanisms in NLP and prior work in XCiT [4]. A discussion of the differences between CCA and these methods, alongside proper citations, would clarify the novelty and address potential ethical concerns regarding prior art.

5. While SCHEME’s performance is strong and well-motivated, the contribution could be strengthened with deeper analysis. SCHEME integrates Block Diagonal MLP and CCA, elements with roots in existing literature. A more detailed explanation of how SCHEME specifically influences learned representations would underscore its contribution.

6. The comparison methods are outdated and do not represent state-of-the-art accuracy-efficiency models. Including comparisons with recent approaches like FasterViT [5], FastViT [6], and MobileOne [7] would enhance the experimental validity.

7. SCHEME is tested only on small model scales (<60M), which aligns with its efficiency-focused motivation. However, a benchmark of on-device speed on mobile devices is necessary to validate the proposed method’s real-world application, as many recent studies have done [6,7].



[1] Liu Z, Mao H, Wu C Y, et al. A convnet for the 2020s[C]//Proceedings of the IEEE/CVF conference on computer vision and pattern recognition. 2022: 11976-11986.

[2] Chen J, Kao S, He H, et al. Run, don't walk: chasing higher FLOPS for faster neural networks[C]//Proceedings of the IEEE/CVF conference on computer vision and pattern recognition. 2023: 12021-12031.

[3] Yu W, Zhou P, Yan S, et al. Inceptionnext: When inception meets convnext[C]//Proceedings of the IEEE/CVF Conference on Computer Vision and Pattern Recognition. 2024: 5672-5683.

[4] Ali A, Touvron H, Caron M, et al. Xcit: Cross-covariance image transformers[J]. Advances in neural information processing systems, 2021, 34: 20014-20027.

[5] Hatamizadeh A, Heinrich G, Yin H, et al. Fastervit: Fast vision transformers with hierarchical attention[J]. arXiv preprint arXiv:2306.06189, 2023.

[6] Vasu P K A, Gabriel J, Zhu J, et al. FastViT: A fast hybrid vision transformer using structural reparameterization[C]//Proceedings of the IEEE/CVF International Conference on Computer Vision. 2023: 5785-5795.

[7] Vasu P K A, Gabriel J, Zhu J, et al. Mobileone: An improved one millisecond mobile backbone[C]//Proceedings of the IEEE/CVF conference on computer vision and pattern recognition. 2023: 7907-7917.

**Questions:**

Please first deal with the major concerns above. There are some minor issues below.

1. The learnable weight $\alpha$ enables CCA as a regularizer during training, discarded for efficiency during inference. However, this approach may create potential conflicts in optimization objectives. Why not consider using more established techniques, such as Structural Re-parameterization, to achieve structure changes between training and inference?

2. Clarification is needed on the smoothing factor $\tau$ — is it a learnable parameter or a constant?

3. There are a few typographical errors, such as a double 'of' in Line 041 of the abstract.

---

> ### Author Response · Authors · 2024-11-26
>
> We thank the reviewer for their comments and useful feedback.
>
> **$\textbf{W1}$**: We apologize for the formatting issues caused by the small font size (due to the large number of comparisons presented). In the revised version, we have updated the figures with larger font sizes, and we have also included the zoomed-in versions in Figures 5 and 6. These updates will also be incorporated in the final version.
>
>
> **$\textbf{W2}$**: Thank you for your comment. Our approach is motivated based on the observation for ViTs  whose performance being highly sensitive to the expansion ratios in MLP/FFN blocks, making them ideal candidates for evaluating SCHEME. Since our proposed model builds on the Metaformer architecture, we designed and evaluated a family of models using this baseline to ensure a thorough comparison.
>
> To address the reviewer’s concern, we have expanded our analysis to include convolution-based models such as CAFormer, which replaces attention with convolution, and CoAtNet, a hybrid model combining convolution and attention. As demonstrated in Table 10 of the appendix, the SCHEME module consistently improves the accuracy-computation tradeoff for both models, delivering higher throughput with fewer FLOPs and parameters for the same accuracy, thereby validating its broader applicability beyond ViTs.
>
> **$\textbf{W3}$**: Section 4.1 (L361) outlines the hardware specifications and benchmark configurations used for all latency and throughput analyses, performed at the standard 224x224 input resolution. To ensure fair comparisons, all models were tested under identical configuration settings and in the same environment.
>
> **$\textbf{W4}$**: Thank you for the suggestion. We have expanded the discussion of the differences between CCA and XCiT in the related work section (L180-190) for clarity. Below is a summary of the key distinctions.
>
> XCiT utilizes a cross-covariance attention (XCA) operator, which is conceptually similar to a "transposed" version of self-attention and operates across feature channels. The XCiT architecture consists of three main components: the XCA operation, a local patch interaction module, and a feedforward network. The XCA mechanism computes a covariance operation across different head groups, analogous to multi-head attention.
>
> In contrast, we introduce Channel Covariance Attention (CCA) block, which facilitates feature mixing across different channel groups. Unlike XCiT, CCA does not use "heads" or projection matrices for queries, keys, and values. Instead, it leverages the full feature set to directly compute covariance, allowing for a more holistic representation of inter-feature interactions.
>
> Regarding the comparison with linear attention mechanisms in NLP, we would like to clarify that CCA differs in that it retains the use of softmax across the channel features, resulting in a quadratic complexity with respect to the feature dimension. Linear attention, on the other hand, reduces the complexity from O(N^2) to O(N) by expressing self-attention as a linear dot product of kernel feature maps, utilizing the associativity property of matrix products, where N is the token length.
>
> **$\textbf{W5}$**:  We have provided a qualitative analysis of the feature representations in Figures 9 and 10, which illustrate that the model incorporating CCA focuses more effectively on the regions where the target class is present, demonstrating the effectiveness of the proposed method. Additionally, the quantitative ablation studies presented in Tables 9 and 18 show that the feature clusters formed by CCA learn a more diverse set of features that complement each other, leading to a 1% improvement in ensemble accuracy. This further highlights SCHEME’s influence on representation learning.
>
>
> **$\textbf{W6}$**:  We included Table 10 in the appendix that shows the performance of SCHEME across 10 different ViT models **including the latest variant BiFormer**. Unfortunately due to the limited computational resources available in our lab, we are unable to include the comparisons to other ViT variants mentioned by the reviewer. We will include them in the camera-ready version after acceptance. We believe the proposed SCHEME module will generalize effectively to the latest variants, as demonstrated by its performance across 10 different ViTs in Table 10.
>
> [1] Lei Zhu, Xinjiang Wang, Zhanghan Ke, Wayne Zhang, and Rynson Lau BiFormer: Vision Transformer with Bi-Level Routing Attention, CVPR 2023

---

> > ### Author Response · Authors · 2024-11-26
> >
> > **$\textbf{W7}$**:  We would like to clarify that our SCHEME module is composed of standard matrix multiplication operations which is generic and not tied to mobile models. Therefore, it does not introduce any additional computations from its implementation. Since we do not have a mobile platform to perform evaluation, we report the inference speed on an Intel(R) Xeon(R) CPU E5-2630 v4 @ 2.20GHz. As shown in the table below, SCHEME has comparable parameters and FLOPs to the baseline Metaformer while obtaining 0.8% improvement while maintaining the same throughput similar to the GPU environment. Therefore, we believe that our SCHEME can be effectively exported to mobile formats with further optimizations.
> >
> > | Method           | Params (M) | FLOPs (G) | CPU Throughput (images/second) | IN-1K Acc. (%) |
> > |-------------------|--------|-------|----------|----------------|
> > | Metaformer-PPAA-11-e2-S12     | 11.8  | 1.8  |    30 | 78.9          |
> > | SCHEMEformer-PPAA-44-e8-S12 | 11.8   | 1.8  |   30   | 79.7           |
> >
> >
> > Q1: Thank you for this insightful suggestion. We initially explored using structural re-parameterization methods for feature mixing as an alternative to CCA. However, these approaches did not yield significant improvements in performance, which led us to adopt the more cost-effective CCA design.
> >
> > Q2: $\tau$ is a constant and is fixed to one in all our experiments.
> >
> > Q3: Thank you for pointing out the typo, we fixed it in the revision.

---

> > > ### Author Response · Authors · 2024-12-02
> > > **Request for feedback on Rebuttal**
> > >
> > > We sincerely thank you for the feedback on our work. We have carefully considered and addressed the concerns raised in our rebuttal submission. With the rebuttal deadline approaching, we kindly request your feedback on the rebuttal to ensure that it satisfactorily addresses all your comments and any outstanding issues. Please let us know if there are any additional questions or suggestions.
> > >
> > > Thanks,
> > > Authors

---

### Official Review · Reviewer_Ezua · 2024-11-01

**Soundness:** 3
**Presentation:** 3
**Contribution:** 3
**Rating:** 6
**Confidence:** 4

**Summary:**

This paper study the channel mixer of ViT MLPs, showing that dense feature mixing can be replaced by sparse feature mixing of higher internal feature dimensionality for improved accuracy, without increased complexity.  The work presents a range of experiments that sufficiently support its claims.

**Strengths:**

1. The writing is easy to read and clearly explains everything in the paper.
2. The experimental result is good compared to the previous works. Empirically, the method seems to offer strong accuracy, compared to existing methods with similar architectures.

**Weaknesses:**

1. I am concerned about the generalization of this method to other data sets.
2. Some details are missing. For Block Diagonal MLP, how to split the feature vectors of (1)-(2) into disjoint groups? What criteria is it based on？What's the motivation for designing the Channel Covariance Attention (CCA)? It didn't show more details.
3. How is the design like channel mixer relate to improving performance for example? It seems inadequate that none of were seriously discussed in the manuscript.

**Questions:**

1. I am concerned about the generalization of this method to other data sets.
2. Some details are missing. For Block Diagonal MLP, how to split the feature vectors of (1)-(2) into disjoint groups? What criteria is it based on？What's the motivation for designing the Channel Covariance Attention (CCA)? It didn't show more details.
3. How is the design like channel mixer relate to improving performance for example? It seems inadequate that none of were seriously discussed in the manuscript.

---

> ### Author Response · Authors · 2024-11-26
>
> We thank the reviewer for their comments and useful feedback.
>
> **$\textbf{W1}$**: In our work, we demonstrated that SCHEME effectively improves the accuracy-latency trade-off across 10 different ViTs, as summarized in Table 10. Additionally, we showed its improvements in Object Detection and Semantic Segmentation tasks using datasets distinct from ImageNet, as detailed in Tables 5 and 6. These results collectively show the generalizability of SCHEME across diverse models, datasets, and tasks, aligning with standard benchmarks commonly used in the ViT literature.
>
> **$\textbf{W2}$**: As outlined in Equations 3 and 4, SCHEME employs two hyperparameters, \( g_1 \) and \( g_2 \), to define the group structure. These parameters can be adjusted to balance computational complexity and performance. In Table 4, we presented results for configurations with \( g_1 = 4, g_2 = 4 \) and \( g_1 = 1, g_2 = 2 \), chosen to ensure comparable complexity to existing ViTs. Both configurations demonstrated improved accuracy, confirming the effectiveness of our approach. For future work, we plan to incorporate Neural Architecture Search to find the optimal architecture based on the parameter-FLOPs trade-off which is beyond the scope of the current work.
>
> As explained in Section 3.2 (L229), splitting the feature vectors into disjoint groups without inter-group communication reduces accuracy, as shown in Table 7, where the model without CCA achieves 0.6% lower accuracy. This highlights the importance of feature mixing for achieving high performance.
>
> **$\textbf{W3}$**: Regarding the motivation for CCA, as discussed in Section 4.2 (L428, L448), its design has a regularizing effect that promotes the formation of feature clusters within transformer blocks (Tables 9 and 18, Figures 9 and 10), leading to improved accuracy. We also addressed the impact of the mixing weight $\alpha$ in response to reviewer N5s3, further supporting the benefits of CCA in enhancing model performance.

---

> > ### Comment · Reviewer_Ezua · 2024-11-28
> >
> > Thanks for authors' rebuttal, which has addressed my concern about this paper.

---

### Official Review · Reviewer_N5s3 · 2024-11-02

**Soundness:** 2
**Presentation:** 1
**Contribution:** 2
**Rating:** 3
**Confidence:** 4

**Summary:**

The paper proposes SCHEME (Scalable CHannEl MixEr), a channel mixing mechanism for Vision Transformers (ViTs). The method focuses on replacing the dense MLP layers in the transformer’s channel mixing block with a block diagonal MLP (BD-MLP) structure, allowing for larger expansion ratios and improving computational efficiency. Additionally, the authors introduce a Channel Covariance Attention (CCA) mechanism to enable inter-group communication, which is discarded after training to reduce inference complexity. The SCHEME mechanism is integrated into multiple Vision Transformer architectures and is evaluated across various benchmarks, showing improved accuracy while maintaining efficiency.

**Strengths:**

1. SCHEME introduces a clever way of reducing complexity by leveraging block diagonal MLPs, which directly targets the computational bottleneck of standard transformers.
2. The addition of the Channel Covariance Attention (CCA) during training adds flexibility and improves feature clustering without increasing inference complexity, making it an efficient regularization tool.
3. The approach is shown to be effective across multiple transformer backbones (MetaFormer, T2T, Swin), demonstrating its adaptability in various settings.

**Weaknesses:**

1. The manuscript contains several grammatical errors and formatting issues, such as the phrase 'In result' in the abstract, as well as figures that are too small to clearly convey the details, which fails to meet the standards of ICLR.
2. Experiments focuses heavily on comparisons with a few models like MetaFormer, lacking comparisons with state-of-the-art transformer-based models like PVT, UniFormer or TransNeXt.

**Questions:**

1. What is the interpretation of the gradual decay of the mixing weight \((1 - \alpha)\) over training epochs, and how does it affect model performance or convergence?

---

> ### Author Response · Authors · 2024-11-26
>
> We thank the reviewer for their comments and useful feedback.
>
> **$\textbf{W1}$**: We apologize for the grammatical errors and formatting issues caused by the small font size (due to the large number of comparisons presented). In the revised version, we have fixed the errors, increased the font size in the figures, and we have also included the zoomed-in versions in Figures 5 and 6. These updates will also be incorporated in the final version.
>
> **$\textbf{W2}$**: Table 10 in the appendix demonstrates the improvement achieved by the SCHEME module across 10 different ViTs. We have expanded the comparison to include three additional models, including original and widely used ViTs such as ViT, DeiT, and the latest ViT variant, BiFormer [1]. The table clearly shows that the SCHEME module consistently enhances accuracy (**upto 1.5%**) across all models while either maintaining or improving the image throughput.
>
> Our proposed model is based on the Metaformer, and we introduce a family of models built on this baseline, comparing them extensively against it. Additionally, we compare our approach with prominent and influential ViTs in the literature, such as Swin, CS-Win, and DaViT, as shown in Table 10. However, given the large number of Vision Transformer variants in the literature (e.g., PVT, Uniformer) and the computational budget constraints for running these models, we could not include more methods in our comparison. We will include them in the revised version after acceptance. Nonetheless, we believe the proposed SCHEME module will generalize effectively to the latest variants, as demonstrated by its performance across 10 different ViTs in Table 10.
>
>
> $\textit{What is the interpretation of the gradual decay of the mixing weight ((1 - $\alpha$)) over training epochs, and how does it affect model performance or convergence?}$
>
> The gradual decay of the mixing weight $(1 - \alpha)$ over training epochs reflects the model's shift from relying heavily on inter-group communication (enabled by the mixing) to prioritizing the independent learning of feature clusters within each group. This facilitates the emergence of more diverse and complementary features within the groups, improving the model's overall performance. We show that Channel Covariance Attention (CCA) promotes the clustering of features into naturally independent groups, improving class separability, as demonstrated in Tables 9 and 18. Experimental results show that models trained with CCA achieve higher ensemble accuracy (+1.09%/+0.6%) and exhibit superior class separability. Additionally, Table 7 highlights the accuracy improvements achieved with CCA, while Table 8 demonstrates that alternative methods are less effective compared to CCA.
>
> [1] Lei Zhu, Xinjiang Wang, Zhanghan Ke, Wayne Zhang, and Rynson Lau BiFormer: Vision Transformer with Bi-Level Routing Attention, CVPR 2023

---

> > ### Comment · Reviewer_N5s3 · 2024-11-28
> >
> > Thank you for the authors’ rebuttal, which addresses some of my earlier concerns. However, issues remain regarding the paper’s quality and the lack of sufficient method comparisons. I therefore maintain my rating.

---

> ### Author Response · Authors · 2024-11-29
>
> We thank the reviewer for their feedback on our rebuttal.
>
> To improve readability and address quality issues, we have corrected the grammatical errors, increased the font sizes and included zoomed-in versions of the figures in the updated revision. If the reviewer could point out specific areas of the paper that could benefit from further improvements, we will gladly make additional updates and incorporate them in the final revision.
>
> Regarding the request for additional comparisons, we are actively training the models with the backbones suggested by the reviewer and will include these results in the final version after acceptance. As highlighted in Table 10, we have already compared SCHEME with **10 different ViTs** from the literature, demonstrating its effectiveness in **improving the accuracy-latency tradeoff** across a variety of architectures, including both **attention-based and convolution-based** designs. This demonstrates the versatility of SCHEME and its ability to advance the state-of-the-art across various ViT architectures. We hope this addresses the reviewer’s concerns.

---

### Author Response · Authors · 2024-11-26

We sincerely thank all the reviewers for their valuable comments and constructive feedback on the paper. We apologize for the delayed response, as conducting additional experiments took time due to our limited computational resources. The paper has been revised based on the feedback, with changes highlighted in blue for the reviewers' convenience. We hope these updates address the reviewers' concerns.

Thanks,

Authors

---

### Note · Authors · 2025-05-16

I have read and agree with the venue's withdrawal policy on behalf of myself and my co-authors.

---

### Meta-Review · Area_Chair_Jj45 · 2024-12-22

**Metareview:**

The paper proposes a scalable channel mixing mechanism for Vision Transformers (ViTs). Its main components are block diagonal MLP (BD-MLP) and channel covariance attention (CCA). The proposed method is integrated into multiple Vision Transformer architectures and is evaluated across various benchmarks, showing improved accuracy while maintaining efficiency. The main concerns raised by the reviewers were the marginal novelty, the quality of writing, and the limited experimental comparisons and analyses. In particular, all reviewers commonly pointed out the need for more extensive experiments with recent architectures and benchmarks. The authors addressed some of the corners during discussions but failed to address the issue of limited experiments fully. After post-rebuttal discussion, two reviewers remain rejects (one reject and one borderline reject) and while other two reviewers gave accepts (borderlines); there was no strong support from the reviewer. AC agrees with reviewers (in particular, N5s3 and vWvm) that the generalizability of the proposed method needs to be validated on more recent architectures and also larger models, which is lacking in the current version. AC thus recommends rejection.

**Additional Comments On Reviewer Discussion:**

After the authors' rebuttal, the issue of limited experiments and analyses was not assuaged, while the others were relatively well addressed. The authors provided additional experiments with 10 different ViT models, but the results were limited to small models and didn't include the comparisons to other ViT variants mentioned by the reviewer. Two reviewers raised their ratings to borderline accept, but two reviewers remained negative (reject and borderline reject) due to insufficient experimental evidence for generalizability. AC agrees with the negative reviewers that the generalizability of the proposed method needs to be validated on more recent architectures and also larger models.

---

### Decision · Program_Chairs · 2025-01-22

Reject